# Morpho-molecular genetic diversity and population structure analysis in garden pea (*Pisum sativum* L.) genotypes using simple sequence repeat markers

Akhilesh Sharma[1]*, Shimalika Sharma[1], Nimit Kumar[2], Ranbir Singh Rana[3], Parveen Sharma[1], Prabhat Kumar[4], Menisha Rani[1]

1 Department of Vegetable Science & Floriculture, Chaudhary Sarwan Kumar Himachal Pradesh Krishi Vishvavidyalaya, Palampur, Himachal Pradesh, India, 2 Department of Genetics and Plant Breeding, Chaudhary Sarwan Kumar Himachal Pradesh Krishi Vishvavidyalaya, Palampur, Himachal Pradesh, India, 3 Centre for Geo Informatics Research and Training, Chaudhary Sarwan Kumar Himachal Pradesh Krishi Vishvavidyalaya, Palampur, Himachal Pradesh, India, 4 PIU-NAHEP, Krishi Anusandhan Bhavan-II, Indian Council of Agricultural Research, Pusa, New Delhi, India

* assharmaakhil1@gmail.com

**Data Availability Statement:** All relevant data are within the paper and its Supporting Information files.

## Abstract

Garden pea (*Pisum sativum* L.) is a self-pollinated plant species which played an important role for the foundation of modern genetics. Genetic diversity among 56 garden pea genotypes was assessed using 12 morphological descriptors, 19 quantitative traits and 8 simple sequence repeat (SSR) markers. Eight morphological descriptors were found polymorphic, and highest Shannon diversity index was recorded for pod curvature (1.18). Mahalanobis $D^2$ illustrating genetic divergence arranged 56 genotypes into six clusters, with the highest inter-cluster distance between clusters IV and VI (18.09). The average values of Na (number of alleles), Ne (effective number of alleles), I (Shannon's Information index), PIC (polymorphism information content), Ho (observed heterozygosity) and He (expected heterozygosity) were 3.13, 1.85, 0.71, 0.36, 0.002 and 0.41, respectively. Pair wise genetic distance among all pairs of the genotypes varied from 0.33 to 1.00 with an average of 0.76. Based on genetic distance, the genotypes were classified into two main clusters (A and B) by cluster analysis, whereas structure analysis divided the genotypes into four sub-populations. The SSR makers indicated that present of genetic variability among the studied genotypes. When, we compared the groups formed by agro-morphological and molecular data, no genotypes were observed, indicating that both stages of characterization are crucial for a better understanding of the genetic variability. Hybridization between genetically diverse genotypes can be exploited to expend the genetic variability and introduce new traits in the pea breeding program.

**Funding:** The authors are grateful to the National Agricultural Higher Education Project (NAHEP) – Indian Council of Agricultural Research (ICAR), New Delhi for financial support during the study.

**Competing interests:** The authors have declared that no competing interests exist.

## Introduction

Garden pea (*Pisum sativum* L; 2n = 2x = 14), belonging to family Leguminosae is an important cool season vegetable crop grown throughout the world for its tender green pods, seeds and foliage. It is the oldest model object of plant genetics and one of the most agriculturally important legumes in the world [1]. Garden pea is quite palatable and excellent food for human consumption, which is eaten as fresh, canned, frozen and in dehydrated forms [2]. It provides an exceptionally diverse nutrient profile of health building substances like vitamins, minerals and also lysine, a limiting amino acid in cereals [3]. Fresh pea pods are excellent sources of folic acid, ascorbic acid (vitamin-C), ß-sitosterol and vitamin-K [4]. Antibacterial, antidiabetic, antifungal, anti-inflammatory, anti-hypercholesterolemia, antioxidant activities and anticancerous properties further support its dietary benefits [5]. Currently, pea is an important source of food and feed in developing and developed countries, respectively [6]. India holds the number second spot in the world in pea production as well as area [7]. However, local landraces are becoming less profitable to farmers due to the reduction in yield which is affected by various biotic like pea aphid (*Acyrthosiphon pisum*), pea weevil (*Bruchus pisorum* L.), aschochyta blight (*Ascochyta pisi*), powdery mildew (*Erysiphe polygoni*) and abiotic stresses [8, 9]. To overcome the further economic loss in the context of biotic and abiotic stresses, there is a dire need to breed resistant and high yielding varieties. The domesticated *Pisum sativum* has a wide variability with respect to morphological attributes, especially pod number, pod shape, pod length, seed number, pod maturity, and quality traits. Therefore, knowledge about the variability in pea genotypes is essential for successful conservation, protection and its use in breeding program, and also for broadening the genetic basis of cultivated cultivars. Different markers like morphological and molecular can be employed to estimate the genetic diversity. Among them, morphological characterization is the first step in classification and description of genetic resources [10]. Newly developed variety has to undergo registration process for plant variety protection so that other stakeholders can be used it commercially for which DUS test is a pre-requisite. DUS test is based on morphological descriptors only for the varietal registration in India, which poses a limitation as the traditional DUS testing method is resource expensive, time-consuming [11] and character expression is affected by environment due to G × E interaction [12, 13]. However, precise estimation of genetic diversity on the basis of morphological data only cannot be revealed due to limited polymorphism and environmental interference [14]. In addition, molecular markers along with the morphological markers can be an effective and suitable means for the precise estimation of genetic diversity. Meanwhile, information on characterization of garden pea genotypes using combination of agro-morphological approach together with a molecular evaluation is scare. For assessing the genetic diversity among large number of pea germplasm in short time at low costs, the use of simple sequence repeat (SSR) markers is an effective and ideal approach as they are highly reproducible, multi-allelic nature, distribution throughout the genome, co-dominant nature of inheritance and easy detection by PCR which makes it most suitable for whole genome characterization [15, 16]. In addition, SSRs performed superiority than SNPs in resolving population structure [17]. SSR markers were found very effective to estimate the genetic diversity in peas germplasm [18, 19]. Both approaches are addressed in the current research project in order to gain a better knowledge on available garden pea genotypes and design suitable strategies for their future exploitation.

The aim of this study was deciphering the genetic diversity among pea genotypes by employing combined morphological qualitative and quantitative characters along with SSR markers.

## Material and methods

### Plant materials

Fifty-six *Pisum sativum* L. genotypes (S1 Table) collected from diverse ecological areas (CSKHPKV, Palampur; IARI, New Delhi; IIVR, Varanasi under All India Coordinated Research Projects (AICRP); ICAR-RS, Katrain; PAU, Ludhiana and CSAUA&T, Kanpur) of India to elucidate their genetic diversity and population differentiation.

### Phenotyping

All the genotypes were phenotype for 12 DUS descriptors and 19 other morphological and quality traits for two consecutive years (2019–2020, 2020–2021) in a randomized complete block design (RCBD) with three replications. Each genotype was assigned to two rows of 2.5 m length with inter and intra-row spacing of 45 cm and 10 cm, respectively. The soil of experimental field was clay loam with pH 5.7 and the observations were recorded on randomly taken ten plants of each genotype in each replication for the 12 DUS traits as per the guidelines prescribed for DUS test of pea by PPV&FRA, the data of 56 genotypes was converted to the respective DUS test scores and further transformed to binary data. The other morphological and quality traits viz., days to 50% flowering [Ch1], first flower node [Ch2], days to first picking [Ch3], number of branches per plant [Ch4], internodal length (cm) [Ch5], number of nodes per plant [Ch6], plant height (cm) [Ch7], pod length (cm) [Ch8], pod width (cm) [Ch9], seeds per pod [Ch10], shelling (%) [Ch11], number of pods per plant [Ch12], and pod yield per plant (g) [Ch13], average pod weight (g) [Ch14], harvest duration (days) [Ch15], moisture content (%) [Ch16], total soluble solids (˚Brix) [Ch17], ascorbic acid (mg/100g) [Ch18], and sugar content (mg/g) [Ch19] were also estimated. All plants of each genotype were scored for powdery mildew disease reaction at its peak stage prior to seed maturity as described by Sharma et al. [3].

### Genotyping using SSR markers

Genomic DNA which was isolated from the fresh leaves bulked of ten randomly chosen plants per genotype by cetyl trimethyl ammonium bromide (CTAB) method Clarke [20] with certain modifications. Leaves samples grinded by using pestle and mortar and extract were put into 2 ml of centrifuge tube which contain 800 µl of extraction buffer. Tubes were incubated at 65˚C for 40–50 minutes in hot water bath and shaken well after 10 minutes intervals. Then chloroform: isoamyl alcohol (24:1 ratio) were added in tubes (800 µl) and were kept on shaker for 25 minutes and then centrifuged at 13000 rpm for 7 minutes. The supernatant (upper phase) was transferred into 1.5 ml tubes and equal volume of chilled isopropanol (600 µl) was added and kept in -20 freezer for 2 hour or for overnight. Then tube was centrifuged at 13000 rpm for 7 minutes. The supernatant was discarded and the pellets were rinse with 70% alcohol (200 µl) and centrifuged at 7000 rpm for 3 minutes and the pellets were air dried. Nuclease free water (50 µl) or 1 X TAE buffer was added to dissolve the pellet and stored at -20˚C.

**Primer selection.** Pea SSR primer pairs were obtained from the previous papers of Mohamed et al. [21], Sharma et al. [22] and Singh et al. [23]. A total of 50 SSR primer pairs (S2 Table) were screened for polymorphism and eight primers were showed polymorphism which were scored for further analysis and rest of the primers were excluded from the study.

**PCR amplification and band profiling.** The PCR reactions was carried out in 0.2 ml PCR tubes containing 10µl reaction mixture, which contained 1µl of DNA template, 0.6 µl of forward primer, 0.6 µl of reverse primer, 3.5 µl of master mix and 4.3 µl of nuclease free water for each genotype. PCR tubes containing reaction mixture were thoroughly mixed by using spiner and subjected to the PCR amplifications in a DNA Thermal Cycler as follows: an initial

denaturation step for 5 min at 94°C, followed by 32 cycles for 1 min at 94°C, 55–65°C (based on primer temperature) for 30 seconds, 72°C for 30 seconds followed by an extension step of 5 min at 72°C. The PCR products were then stored at 4 °C. The PCR products were subjected to electrophoresis in 2.5% Agarose gel at 100 V for 120–150 min. and amplicon size was estimated using 100 bp DNA ladder, which were under UV using a gel documentation system.

## Data analysis

**DUS data analysis.** Binary data of 12 DUS descriptors was subjected to construct the dendrogram using NTSYS-pc (version 2.02). The diversity index (DI) was calculated for each descriptor using formula: $H = -\sum_{i=1}^{R} Pi \ln Pi$

**Morphological diversity analysis.** Phenotypic data of 19 characters (other than DUS descriptors) was subjected to Mahalanobis $D^2$ statistics [24] followed by grouping into different clusters by Toucher's method as suggested by Rao [25]. For conducting $D^2$ analysis, the computer program WINDOSTAT 8.0 developed by Indostat Services was used. Mahalanobis $D^2$ analysis between two genotypes estimated on the basis of the 'p' characters is given by the equation: $D^2 = \sum_{i=1}^{p} \sum_{j=1}^{p} wij(Xi1 - Xi2)(Xj1 - Xj2)$, Where, wij = variance-covariance matrix, $w^{ij}$ = reciprocal of (wij), (i j = 1,2......, p), $X_{i1}$ = sample mean for $i^{th}$ character for first sample, $X_{i2}$ = sample mean for $i^{th}$ character for second sample. In addition, the grouping pattern of the 56 garden pea genotypes was estimated by principal component analysis (PCA) in software XLSTAT [26] using EIGEN procedure on the basis of correlation coefficient between two genotypes.

**Molecular analysis.** For the estimation of genetic diversity of garden pea genotypes, molecular data of each polymorphic marker was converted into binary format as 1 for presence and 0 for absence at each locus. Similarity matrix were generated using Jaccard's coefficient, $J_{ij} = C_{ij}/(n_i + n_j-c_{ij})$, where '$C_{ij}$' is the number of positive matches between two genotypes, while $n_i$ and $n_j$ is the total number of band in genotype i and j, respectively, in SIMQUAL program of NTSYS–PC package (version 2.02) [27], whereas Genetic distances (GD) were calculated as GD = 1 –[$C_{ij}/(n_i+n_j-C_{ij})$] and dendrogram was constructed using the UPGMA algorithm in SAHN program of NTSYS–PC package (version 2.02). POPGEN computer software version 1.32 was used to calculate various genetic diversity parameters viz., Ho (observed heterozygosity), He (expected heterozygosity), Av. He (average heterozygosity), Ne (effective number of alleles), Na (number of alleles) and I (Shannon's Information index) per locus were estimated Yeh and Boyle [28]. The Polymorphic information content (PIC) were calculated as follows:², where $k$ is the total number of alleles and $p_i$ is the frequency of the $i^{th}$ allele in the set of genotypes investigated [29]. Genetic structure of the 56 garden pea genotypes was determined by following Bayesian model-based program STRUCTURE (v.2.3.3) [30]. The analysis was done with k ranging from 1 to 10 using an admixture model with 10,000 burning periods and 10,000 replicates. The final peak of plotting LnPD values were determine using online web-based Structure Harvester program.

Analysis of molecular variance (AMOVA) and The Patterns of genetic relationship contained in the matrix were visualized by Principal Coordinates Analysis (PCoA) in GenALEx 6.5 [31].

# Results

## Agro-morphological diversity and characterization

Consistent scores for 12 morphological descriptors were observed for all the 56 garden pea genotypes over year which indicates the stability of traits. Among the 56 garden pea genotypes, considerable variation was observed for all the important attributes under study except stem

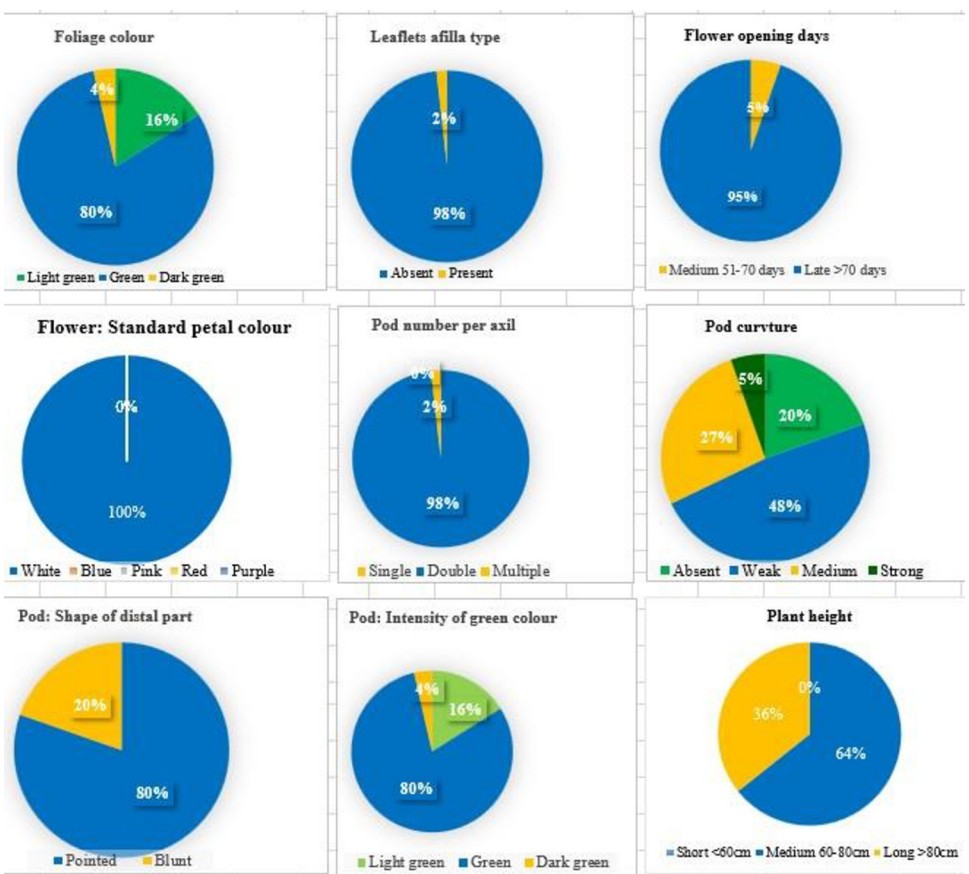

**Fig 1. Frequency of 56 diverse garden pea genotypes based on morphological descriptors (over the years).**

anthocyanin coloration, leaf axil color, stipule type and standard petal color which were found to be monomorphic which were exclude from the analysis (Fig 1). The Shannon diversity index for 12 traits ranged from 0.09 (pod number per axil) to 1.18 (pod curvature) with a mean of 0.49 and a range of 1.09 (Table 1). The similarity coefficient ranged from 0.51 to 1.0 indicating a wide range of genetic variation present in all the genotypes. The cluster analysis grouped all the genotypes into six clusters at 0.81 level of genetic similarity. The grouping pattern of the genotypes under each cluster was mentioned in Fig 2A.

## Mahalanobis D² diversity analysis

The multivariate analysis ($D^2$) illustrating genetic divergence, arranged 56 genotypes into six clusters, cluster I was the largest with 30 genotypes followed by cluster III with 20 genotypes, cluster IV had three genotypes and clusters II, V and VI were mono-genotypic following Tocher's procedure depicted through dendrograms (Fig 2B) indicating vide range of genetic diversity. The inter-cluster distance ranged from 7.96–18.09, the highest inter-cluster genetic divergence was found in clusters IV and VI (18.09) followed by clusters III and VI (14.93), and clusters III and V (14.32) (Table 2).

## Principal component analysis

Principal components were considered significant for eigen values greater than or equal to 1.0. As a result, a total of 77.79% variation was explained by the first seven significant principal

**Table 1. Diversity indices of eight morphological descriptors in garden pea genotypes (over 2 years).**

| Trait | Class or scale of descriptor | Frequency | Relative Frequency (%) | Diversity Index (DI) |
|---|---|---|---|---|
| Foliage colour | Light green | 9 | 16.07 | 0.59 |
| | Green | 45 | 80.35 | |
| | Dark green | 2 | 3.57 | |
| Leaf: Leaflets afila type | Absent | 55 | 98.21 | 0.09 |
| | Present | 1 | 1.78 | |
| Flower opening days | Extra early <40 days | - | - | 0.21 |
| | Early 40–50 days | - | - | |
| | Medium 51–70 days | 3 | 5.35 | |
| | Late >70 days | 53 | 94.64 | |
| Pod number per axil | Single | - | - | 0.09 |
| | Double | 55 | 98.21 | |
| | Multiple | 1 | 1.78 | |
| Pod curvature | Absent | 11 | 19.64 | 1.18 |
| | Weak | 27 | 48.21 | |
| | Medium | 15 | 26.78 | |
| | Strong | 3 | 5.35 | |
| Pod: Shape of distal part | Pointed | 45 | 80.35 | 0.50 |
| | Blunt | 11 | 19.64 | |
| Pod: Intensity of green colour | Light green | 9 | 16.07 | 0.59 |
| | Green | 45 | 80.35 | |
| | Dark green | 2 | 3.57 | |
| Plant height | Short <60cm | - | - | 0.65 |
| | Medium 60-80cm | 36 | 64.28 | |
| | Long >80cm | 20 | 35.71 | |
| Mean | | | | 0.49 |
| Maximum | | | | 1.18 |
| Minimum | | | | 0.09 |
| Range | | | | 1.09 |

components (Table 3). First principal component PC1 described 21.39% of the total variance which was mainly contributed by pod yield per plant followed by PC2 which accounted for 18.17% variation mainly through days to first picking and PC3 which contributed 10.86% variation and was attributed mainly due to pod width. Biplot for PC1 and PC2 also indicated that maximum genotypes were unique as they fall in different corners of biplot. None of the variable fall in the left corner of the lower half of the biplot which indicated that there may be some variables which were involved in variance but did not take into consideration (Fig 3).

### Diversity analysis at molecular level

**Genetic diversity and population structure analysis.** SSR primers were used for molecular marker analysis among the 56 garden pea genotypes (Fig 4). The parameter values of number of alleles per locus (Na), effective number of alleles per locus (Ne), Shannon's Information index (I), polymorphism information content (PIC), Observed heterozygosity, expected heterozygosity and average heterozygosity per locus were used to estimate genetic diversity (Table 4). SSR produced amplification products in size range of 150–480 bp, makers detected a total of 25 alleles which varied from 2 to 4 with an average of 3.13 alleles per locus. However, the effective number of alleles per locus were ranging from 1.16 to 3.09 with a mean of 1.85.

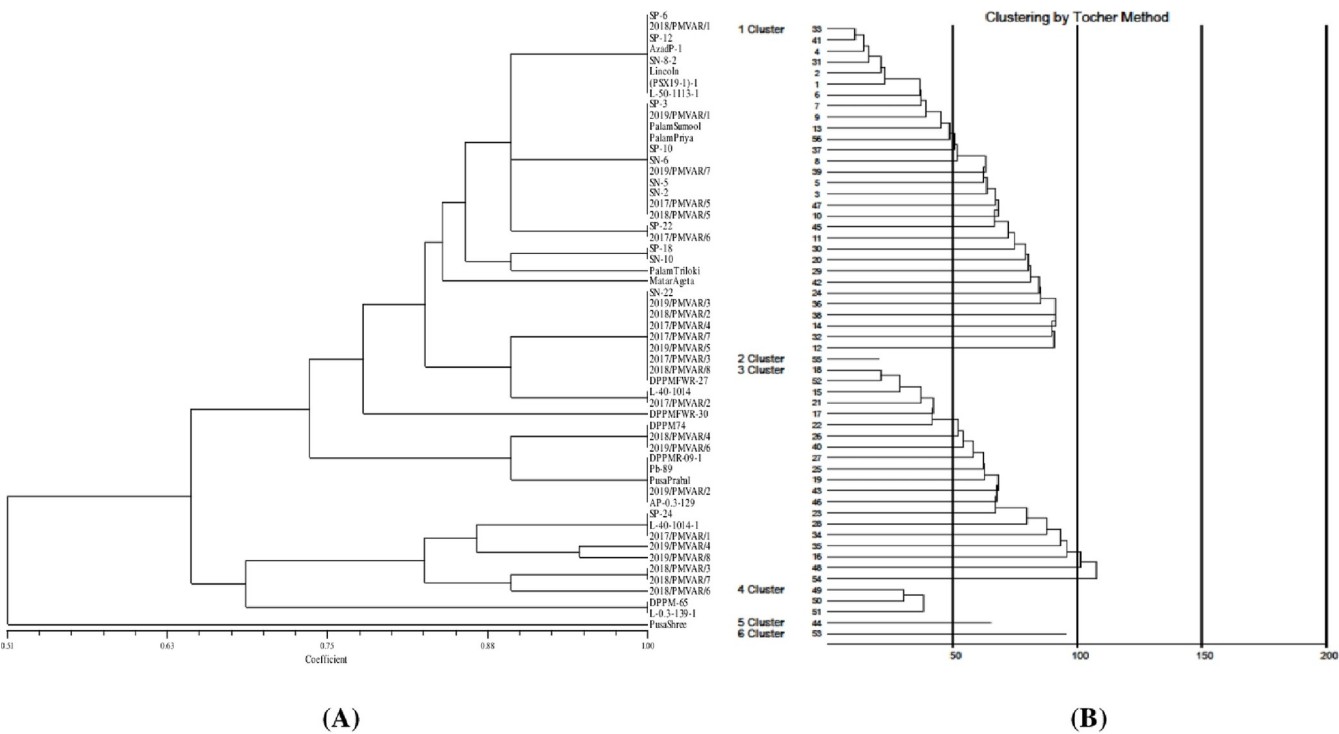

**Fig 2.** Dendrogram of 56 garden pea genotypes on (A) morphological descriptors using squared Euclidean distance (B) 19 morphological traits using Mahalanobis $D^2$ clustering (pooled analysis).

The mean value of Shannon's Information index (I) was 0.71, with a range of values from 0.26 to 1.21. The polymorphic information content (PIC) was varied from 0.13 to 0.61 with an average value of 0.36. Observed and expected heterozygosity ranged from 0.00 to 0.02 and from 0.14 to 0.68, respectively. Mean observed and expected heterozygosity were 0.002 and 0.409, respectively. The genetic distance was calculated among fifty-six garden pea genotypes based on SSR markers. The pair wise genetic distance varied from 0.33 (between three pairs) to 1.00 (between 47 pairs) with the mean value of 0.76 (S3 Table). To estimate the relationships and genetic diversity among these garden pea genotypes, cluster analysis using UPGMA algorithm was performed. The genotypes were grouped into two major clusters A and B (Fig 5). Cluster A comprised of 53 genotypes, whereas three genotypes (L-40-1014-1, DPPMR-09-1 and L-40-1014) were placed in cluster B. Cluster A was further divided in to five sub-clusters *viz.*, $A_1$, $A_2$, $A_3$, $A_4$ and $A_5$. The genetic similarity coefficient among the genotypes ranged from 0.60 to 1.00 with an average of 0.80. The sub-cluster $A_1$ comprised of six genotypes *viz.*, (PS×19–1)-1, Palam Sumool, Palam Triloki, Matar Ageta, 2018/PMVAR/6 and 2019/PMVAR/1, $A_2$ comprised of 20 genotypes *viz.*, L-0.3-139-1, SP-24, 2019/PMVAR/7, 2019/PMVAR/4, L-50-1113-

**Table 2. Mahalanobis distance (inter-cluster) between cluster groups of garden pea genotypes (pooled analysis).**

| Clusters | II | III | IV | V | VI |
|---|---|---|---|---|---|
| I | 10 | 12.26 | 13.46 | 11.27 | 12.43 |
| II | | 7.96 | 12.1 | 11.2 | 11.51 |
| III | | | 12.88 | 14.32 | 14.93 |
| IV | | | | 12.44 | 18.09 |
| V | | | | | 9.76 |

**Table 3. PCA of 19 morphological traits with eigenvalues, variability and cumulative variances (pooled).**

| | PC1 | PC2 | PC3 | PC4 | PC5 | PC6 | PC7 |
|---|---|---|---|---|---|---|---|
| Days to 50% flowering | -0.065 | 0.462 | -0.073 | -0.117 | -0.140 | 0.064 | -0.035 |
| First Flower Node | -0.124 | 0.425 | -0.042 | -0.042 | 0.084 | 0.084 | 0.005 |
| Days to first picking | -0.043 | 0.494 | 0.030 | -0.063 | -0.062 | 0.054 | -0.095 |
| Number of branches | 0.227 | 0.105 | -0.108 | 0.535 | 0.047 | 0.118 | 0.327 |
| Internodal length (cm) | 0.181 | 0.183 | 0.140 | 0.021 | 0.482 | -0.356 | 0.061 |
| Nodes per plant | 0.234 | 0.261 | -0.079 | 0.429 | 0.079 | 0.134 | 0.244 |
| Plant height(cm) | 0.049 | 0.231 | -0.053 | -0.181 | 0.533 | -0.266 | 0.057 |
| Pod length(cm) | 0.375 | 0.109 | 0.146 | -0.261 | -0.107 | -0.092 | -0.091 |
| Pod width(cm) | 0.112 | -0.028 | 0.561 | 0.089 | 0.104 | 0.192 | -0.008 |
| Seeds per pod | 0.410 | 0.006 | -0.094 | -0.277 | -0.058 | -0.035 | -0.083 |
| Shelling (%) | 0.210 | 0.015 | -0.367 | 0.100 | -0.019 | 0.298 | -0.349 |
| Pods per plant | 0.262 | -0.027 | -0.454 | -0.063 | 0.035 | 0.009 | -0.078 |
| Pod yield per plant(g) | 0.451 | -0.039 | -0.115 | -0.107 | -0.033 | 0.023 | -0.071 |
| Average pod weight(g) | 0.366 | -0.013 | 0.399 | -0.071 | -0.105 | 0.019 | 0.022 |
| Harvest duration (days) | 0.185 | -0.368 | -0.068 | 0.035 | 0.101 | -0.146 | 0.286 |
| Moisture content (%) | -0.136 | -0.173 | -0.184 | -0.106 | 0.471 | 0.255 | 0.089 |
| Total soluble solids (˚Brix) | 0.082 | 0.042 | 0.113 | 0.441 | -0.106 | -0.155 | -0.284 |
| Ascorbic acid (mg) | 0.079 | 0.007 | 0.174 | -0.196 | 0.184 | 0.710 | 0.132 |
| Sugar content (mg/g) | 0.011 | 0.118 | -0.110 | -0.220 | -0.359 | -0.073 | 0.692 |
| Eigenvalue | 4.066 | 3.452 | 2.065 | 1.546 | 1.370 | 1.230 | 1.052 |
| Variability (%) | 21.398 | 18.171 | 10.869 | 8.136 | 7.210 | 6.475 | 5.535 |
| Cumulative % | 21.398 | 39.569 | 50.439 | 58.575 | 65.785 | 72.260 | 77.795 |

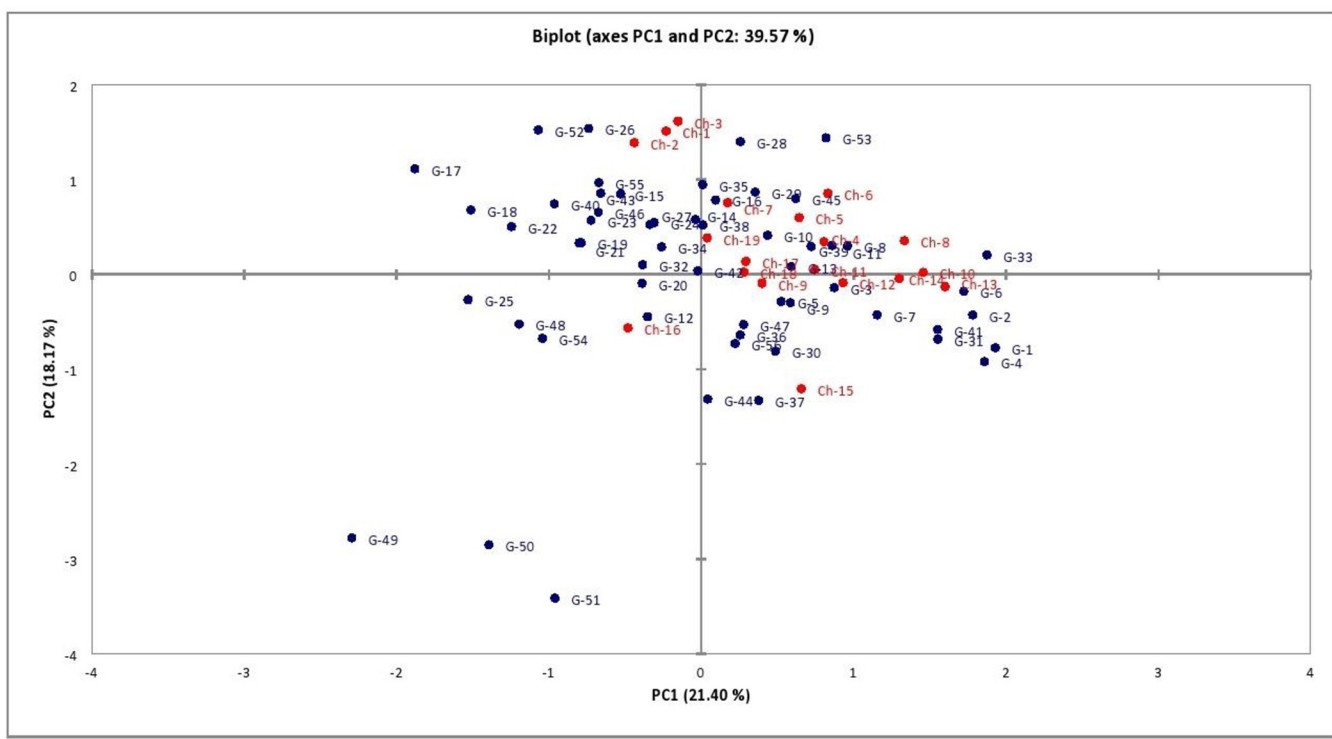

**Fig 3. Principal component analysis biplot for garden pea genotypes (pooled).**

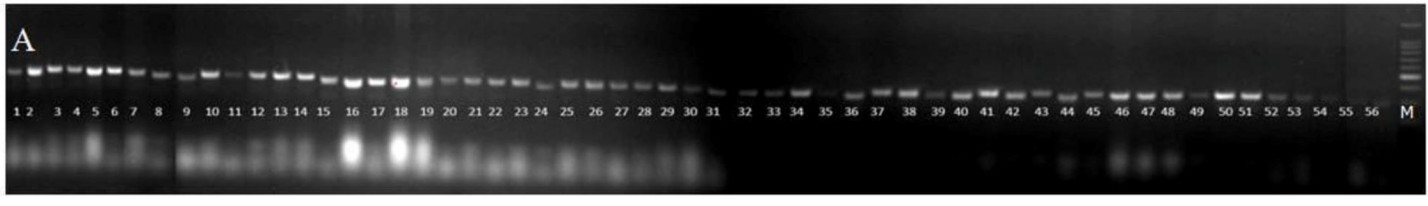

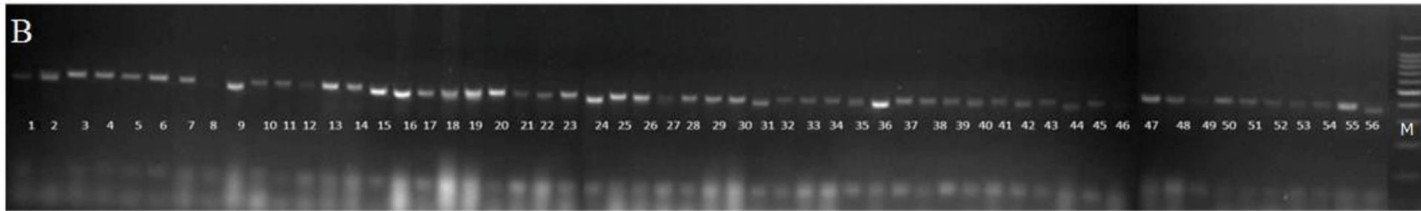

**Fig 4.** Banding patterns of 56 garden pea genotypes amplified with markers (A, AB68) and (B, AA92) on agarose gel. M = 100 bp DNA ladder.

1, 2019/PMVAR/5, Pusa Shree, Pusa Prabal, Palam Priya, DPPM-65, SN-22, 2017/PMVAR/5, 2019/PMVAR/2, 2018/PMVAR/1, SP-6, 2017/PMVAR/4, 2018/PMVAR/2, 2019/PMVAR/3, 2017/PMVAR/6 and 2017/PMVAR/7, $A_3$ comprised of 17 genotypes *viz*., DPPM-74, Azad P1, SP-22, Lincoln, 2018/PMVAR/4, 2018/PMVAR/5, Punjab89, 2018/PMVAR/7, SN-5, SP-18, SP-12, SN-6, SN-2, SP-3, SP-10, SN-8-2 and SN-10, $A_4$ comprised of nine genotypes *viz*., AP-0.3–129, 2018/PMVAR/8, 2017/PMVAR/3, 2018/PMVAR/3, 2019/PMVAR/6, DPPMFWR-30, 2019/PMVAR/8, 2017/PMVAR/1 and 2017/PMVAR/2, whereas $A_5$ had only one genotype DPPMFWR-27. Clustering of population into two distinct groups represents the diversity between the populations.

Analysis of molecular variance (AMOVA) was computed to estimate the genetic diversity among the populations and within the populations. AMOVA depicts high proportion of variability within population *i.e.*, 94% of total variation, whereas only 6% genetic variation among population (Table 5). Principal Coordinate Analysis (PCoA) further re-confirmed the genetic relationship as depicted by cluster analysis. The PCoA grouped 56 garden pea genotypes into four different populations (Fig 6) and showed that accessions are more diverse based on the genetic constitution. Using SSR markers, first principal coordinate explained 15.20% of the total variation followed by second coordinate 12.38% variation and further 11.87% in third coordinate. Very little introgression between the gene pools was observed in the dimension 1 versus 2 comparisons.

**Table 4. Genetic diversity statistics for 8 SSR markers across 56 garden pea genotypes (over 2 years).**

| S. No. | Primers | Na | Ne | I | PIC | Ho | He | Fragment size |
|---|---|---|---|---|---|---|---|---|
| 1 | AB68 | 4 | 3.09 | 1.21 | 0.61 | 0.00 | 0.68 | 300–350 |
| 2 | AA92 | 3 | 1.75 | 0.76 | 0.39 | 0.00 | 0.43 | 310–370 |
| 3 | AA339 | 4 | 2.31 | 0.96 | 0.48 | 0.00 | 0.57 | 150–190 |
| 4 | AA369 | 3 | 1.35 | 0.48 | 0.23 | 0.02 | 0.26 | 250–320 |
| 5 | AB40 | 3 | 2.05 | 0.81 | 0.42 | 0.00 | 0.52 | 200–250 |
| 6 | AB45 | 2 | 1.16 | 0.26 | 0.13 | 0.00 | 0.14 | 150–190 |
| 7 | AD174 | 3 | 1.31 | 0.47 | 0.22 | 0.00 | 0.24 | 420–480 |
| 8 | P1109 | 3 | 1.74 | 0.76 | 0.39 | 0.00 | 0.43 | 400–430 |
| Mean | | 3.13 | 1.85 | 0.71 | 0.36 | 0.002 | 0.41 | |
| St. Dev | | 0.64 | 0.64 | 0.30 | | 0.01 | 0.19 | |

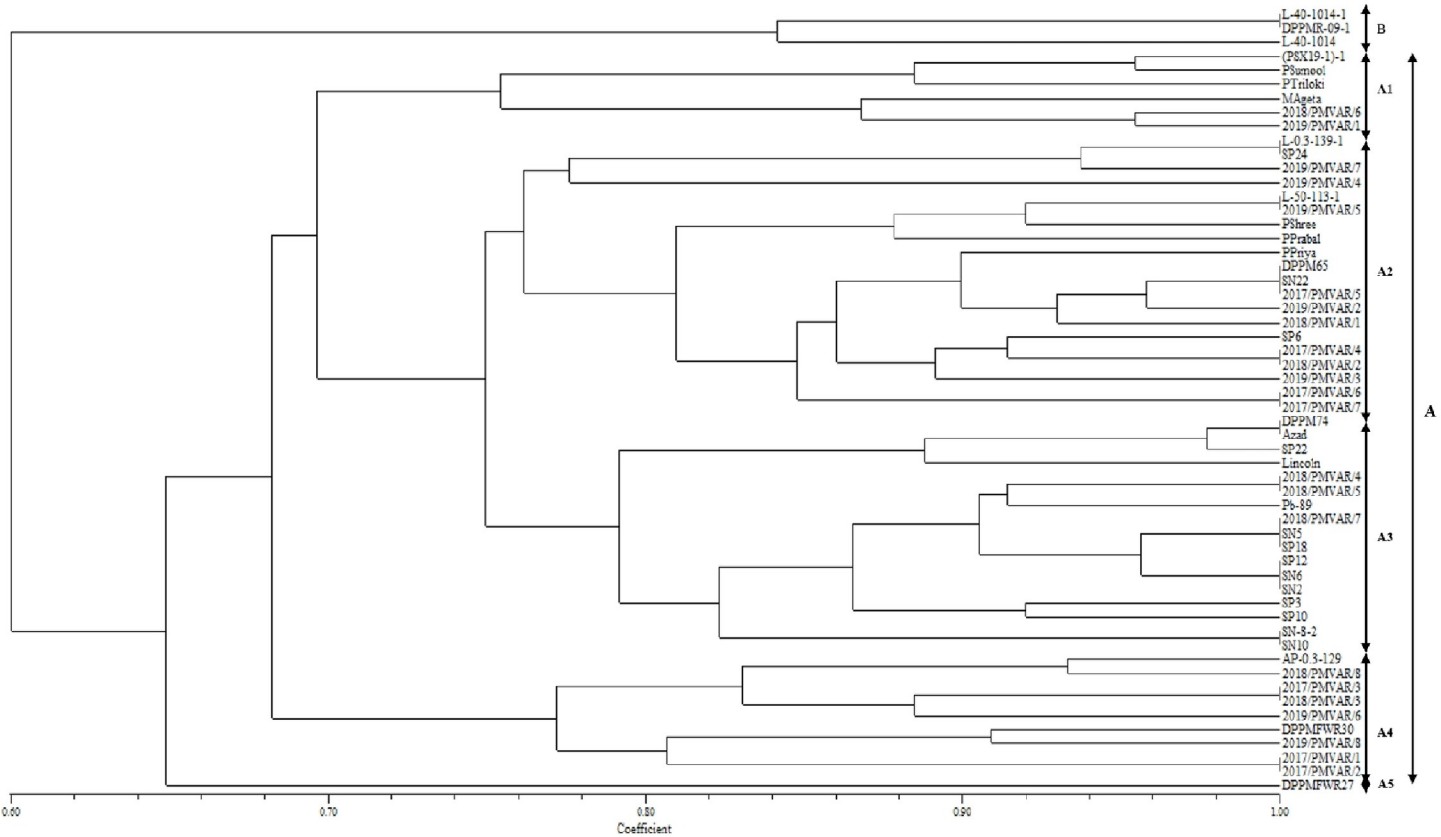

**Fig 5. Dendrogram constructed using genetic distance matrix data calculated from 8 SSR markers among the 56 garden pea genotypes.**

Population structure analysis using SSR markers indicated that the LnP(D) (log-likelihood) increased with the model parameter K value (Fig 7A), further suitable K value (population number) was determined by using the ΔK statistic. The Evanno test revealed that maximum peak value of ΔK was obtained at K = 4 (Fig 7B). Structure analysis indicated that the population of the 56 garden pea genotypes under study was a mixed population having four sub-populations, viz., POP 1, POP 2, POP3 and POP 4 (Fig 8). In total, 14 genotypes (25.0%) were assigned into POP 1, which contained mostly genotypes from CSKHPKV, Palampur, IARI, New Delhi, ICAR-RS, Katrain, and PAU, Ludhiana. POP 2 contained 7 genotypes (12.5%), which were from IIVR, Varanasi under all India coordinated research trial (AICRP) and CSKHPKV, Palampur. POP 3 contained 17 genotypes (30.4%), which mostly were from IIVR Varanasi under AICRP, CSKHPKV Palampur and PAU Ludhiana. POP 4 contained 18 genotypes (32.1%), which were from IIVR, Varanasi under AICRP, CSKHPKV, Palampur, and CSAUA&T, Kanpur. In addition, almost all sub-populations having different pod yield and maturity garden pea genotypes. The results revealed that genetic structure classification

**Table 5. Analysis of genetic differentiation among garden pea genotypes by AMOVA (pooled analysis).**

| Source of Variation | df | SS | MS | Est. Var. | % |
|---|---|---|---|---|---|
| Among Population | 3 | 24.560 | 8.187 | 0.304 | 6% |
| Within Population | 52 | 250.976 | 4.826 | 4.826 | 94% |
| Total | 55 | 275.536 | | 5.130 | 100% |

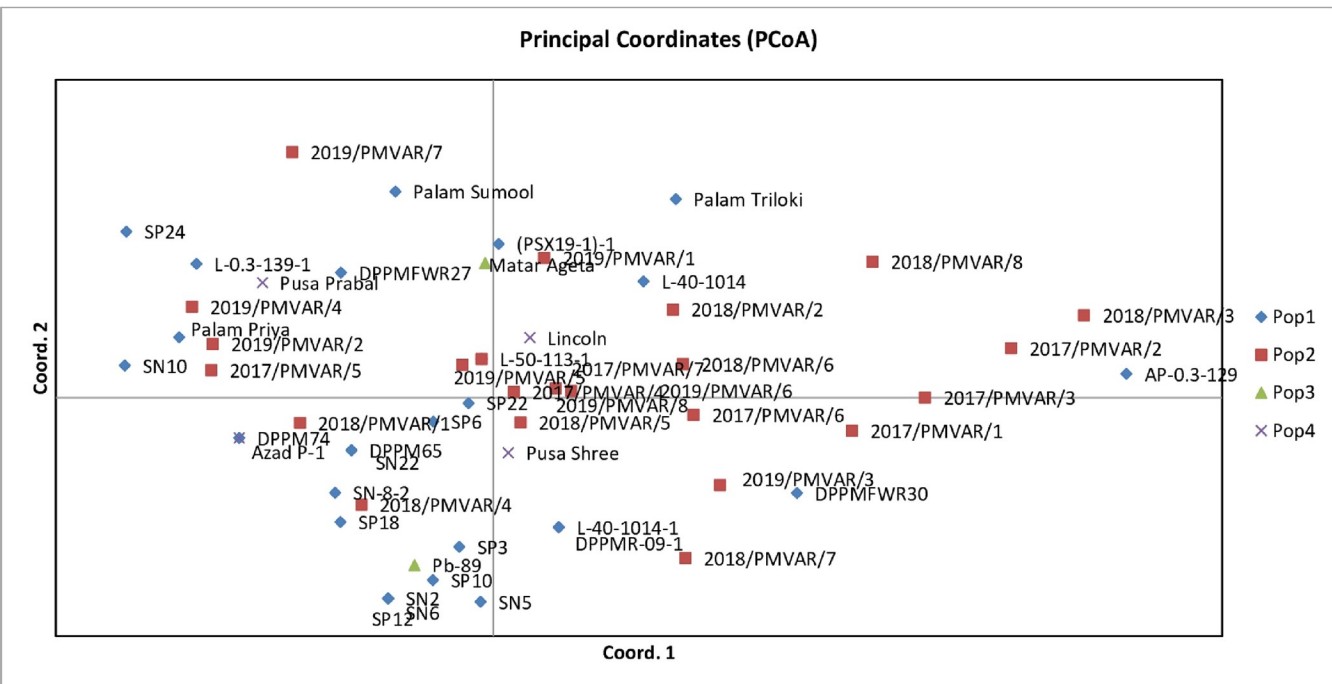

**Fig 6. Principal Coordinate Analysis (PCoA) presenting the genetic diversity of four sub-populations of garden pea at different coordinate.**

pattern of most of the genotypes were not consistent with their geographic origins, maturity and pod yield.

## Comparison of genetic diversity among different populations

The genetic diversity based on different groups among different populations is presented in Table 6. The genetic diversity of the two groups (I and II) evaluated by the SSR markers, among the two groups, group I showed highest genetic diversity based on polymorphism information content (PIC = 0.34), effective number of alleles (Ne = 1.82) and Shannon's

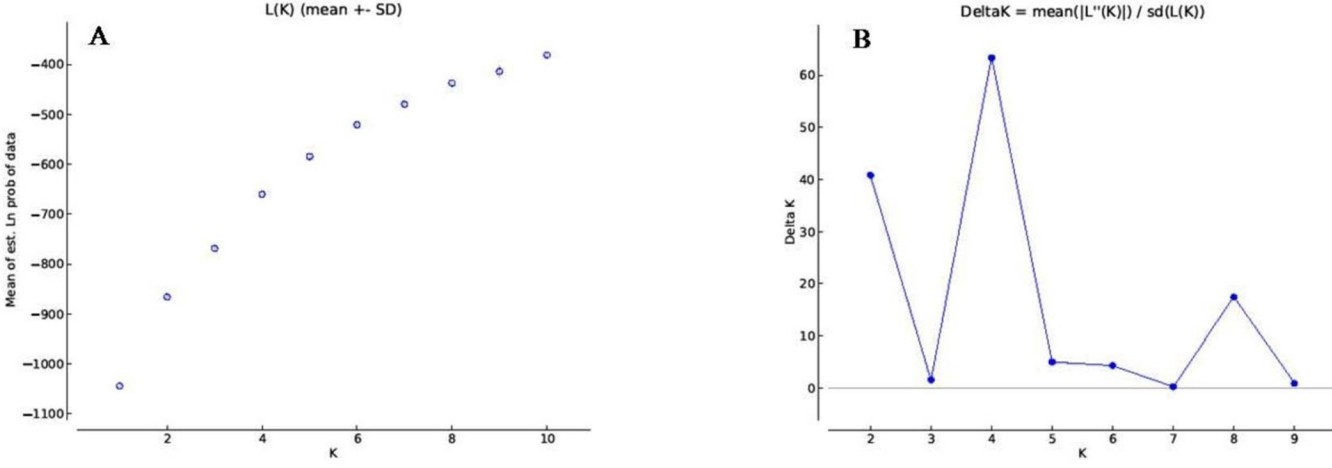

**Fig 7.** Estimated K values (A) and ΔK values (B).

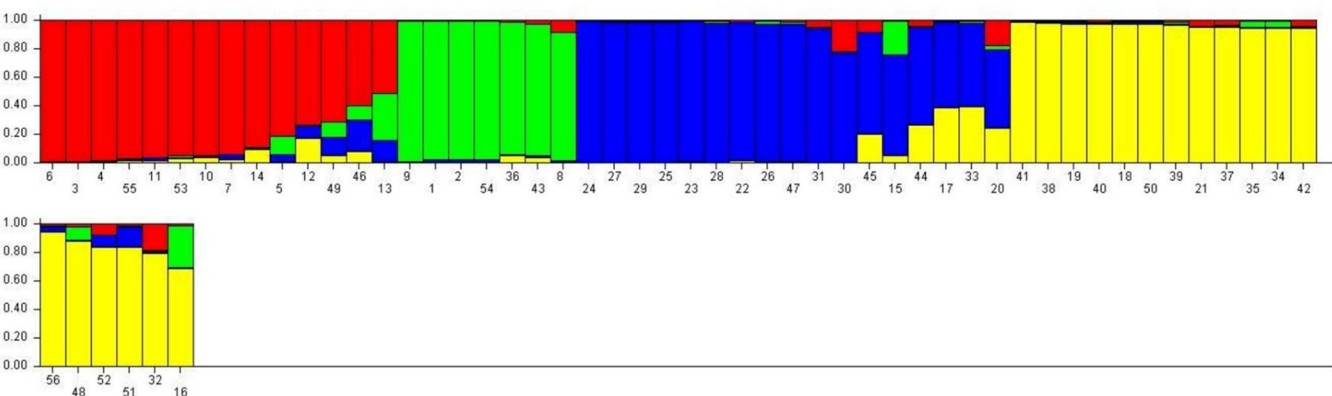

**Fig 8. Population structure of the 56 garden pea genotypes.**

Information index (I = 0.68), whereas group II had the lower genetic diversity based on values of polymorphism information content (PIC = 0.04), effective number of alleles (Ne = 1.10) and Shannon's Information index (I = 0.08). The highest genetic diversity for different maturity populations exhibited by the late maturity population based on polymorphism information content (PIC = 0.35) and number of alleles (Ne = 1.88) and Shannon's Information index (I = 0.67). The genetic diversity among three different yielding populations indicated that low

**Table 6. Group wise genetic diversity statistics for 8 SSR markers across 56 garden pea genotypes (over 2 years).**

| Group/ Population | Na | Ne | I | PIC | Origin and number of genotypes by regions | Total Number of Genotypes |
|---|---|---|---|---|---|---|
| **Grouping by NTSYS** | | | | | | |
| I | 3.00 | 1.82 | 0.68 | 0.34 | CSKHPKV, Palampur (24), IIVR, Varanasi under AICRP (23), IARI, New Delhi (2), PAU, Ludhiana (2), ICAR-RS, Katrain (1), CSAUA&T, Kanpur (1) | 53 |
| II | 1.13 | 1.10 | 0.08 | 0.04 | CSKHPKV, Palampur (3) | 3 |
| **Maturity Group** | | | | | | |
| Early maturity | 2.25 | 1.86 | 0.63 | 0.33 | IIVR, Varanasi under AICRP (5), CSKHPKV, Palampur (1), IARI, New Delhi (1), PAU, Ludhiana (1) | 8 |
| Intermediate maturity | 2.88 | 1.76 | 0.66 | 0.34 | CSKHPKV, Palampur (17), IIVR, Varanasi under AICRP (9), IARI, New Delhi (1), PAU, Ludhiana (1) | 28 |
| Late maturity | 2.63 | 1.88 | 0.67 | 0.35 | CSKHPKV, Palampur (9), IIVR, Varanasi under AICRP (9), ICAR-RS, Katrain (1), CSAUA&T, Kanpur (1) | 20 |
| **Pod Yield Group** | | | | | | |
| High | 2.13 | 1.59 | 0.52 | 0.37 | CSKHPKV, Palampur (9), IIVR, Varanasi under AICRP (4) | 13 |
| Intermediate | 2.13 | 1.54 | 0.46 | 0.24 | IIVR, Varanasi under AICRP (7), CSKHPKV, Palampur (3), PAU, Ludhiana (1) | 11 |
| Low | 3.00 | 1.92 | 0.75 | 0.38 | CSKHPKV, Palampur (15), IIVR, Varanasi under AICRP (12), IARI, New Delhi (2), PAU, Ludhiana (1), ICAR-RS, Katrain (1), CSAUA&T, Kanpur (1) | 32 |
| **Region wise Group** | | | | | | |
| CSKHPKV, Palampur | 2.88 | 1.83 | 0.72 | 0.37 | - | 27 |
| IIVR under AICRP | 2.25 | 1.57 | 0.50 | 0.26 | - | 23 |
| IARI New Delhi | 1.25 | 1.25 | 0.17 | 0.09 | - | 2 |
| PAU Ludhiana | 1.50 | 1.50 | 0.35 | 0.19 | - | 2 |
| ICAR-RS Katrain | 1.00 | 1.00 | 0.00 | - | - | 1 |
| CSAUA&T Kanpur | 1.00 | 1.00 | 0.00 | - | - | 1 |

yielding group had highest genetic diversity with PIC = 0.38, Ne = 1.92 and I = 0.75 followed by high yielding group with PIC = 0.37, Ne = 1.59 and I = 0.52. Compared with the other population types, the intermediate yielding population displayed showed lower genetic diversity based on PIC = 0.24, Ne = 1.54 and I = 0.46. CSKHPKV, Palampur, IIVR, Varanasi under AICRP, IARI, New Delhi, PAU, Ludhiana, ICAR-RS, Katrain and CSAUA&T, Kanpur are the important areas with respect to pea breeding programs. To understand the genetic variation among the garden pea genotypes derived from these areas, the estimation of genetic diversity was carried out. The genotypes derived from CSKHPKV, Palampur had relatively high genetic diversity based on polymorphism information content (PIC = 0.37), effective number of alleles (Ne = 1.83) and Shannon's Information index (I = 0.72), whereas the IIVR, Varanasi under AICRP had the second highest genetic diversity based on polymorphism information content (PIC = 0.26), effective number of alleles (Ne = 1.57) and Shannon's Information index (I = 0.50), whereas the genotypes derived from other areas showed relatively low genetic diversity.

## Discussion

Crop improvement through breeding programmes is depends upon the genetic diversity and population structure of the genetic resource, a wide range of the genotypes of different vegetable crops was studied for the diversity analysis [32]. The genetic base of the local cultivar can be broadened by the plant breeder through the knowledge of genetic diversity. Therefore, estimation of genetic diversity among the genotypes has become an important approach for identifying superior genetically divergent parents along with desirable characters [33]. Morphological characters have been used for the estimation of genetic diversity and relationships among garden pea genotypes from many years. Morphological characters are largely influenced by the environmental factors. A combined approach based on molecular and conventional studies can give a better understanding of variation pattern among the genetic resource that can be exploit to broadening the genetic base for important characters [34]. Population size and type of molecular marker are also the important factors that influence the estimation of population structure and genetic diversity. SSR markers are the important molecular markers to establishment of relationships and genetic diversity as they are polymorphic, highly reproducible, co-dominant in nature and abundant in plant genomes [35, 36].

Fifty-six genotypes of garden pea collected from different regions of India were subjected to morpho-molecular genetic diversity. Morphological descriptors indicated the variation among the genotypes with highest Shannon diversity index for pod curvature (1.18) and grouped the genotypes in six clusters at 0.81 level of genetic similarity. Morphological descriptors can be effectively used for identification and grouping of genotypes which can be used in hybridization program for the development of improved varieties. Similar studies were also performed by Singh et al. [37], they grouped 35 varieties of vegetable pea on the basis of 19 descriptors into different categories for each character. On the basis of $D^2$ analysis using 19 morphological traits, genotypes were arranged into 6 clusters following Tocher's procedure [25] and also depicted through dendrograms (Fig 2B). Different clustering patterns in garden pea genotypes were also reported by earlier workers [38–40]. The greatest inter-cluster distance was observed between cluster IV (Pusa Shree, Matar Ageta-6, Palam Triloki) and VI (Palam Sumool), indicated that the genotypes belonging to these clusters were genetically diverse and could offer relatively better parental lines; the progenies of these genotypes can be useful in further breeding programs for obtaining a wide spectrum of variation. On the other hand, the crosses involving the diverse genotypes would be expected to manifest maximum heterosis and are more likely to evolve desirable recombinants in segregating generations. Among the above

mentioned genotypes of cluster IV, genotype namely 'Pusa Shree' was found moderately resistant (MR) against powdery mildew disease (field screening), genotype 'Palam Triloki' was found moderately susceptible (MS), while genotype 'Mater Ageta-6' was susceptible to powdery mildew disease of pea (S4 Table). The genotype of cluster VI namely 'Palam Sumool' was resistant against powdery mildew disease (S4 Table). Hence, the diverse genotype Matar Ageta-6 (S) and Palam Sumool (R) can be used to generate the mapping population for powdery mildew disease in garden pea. Based on inter-cluster distance, the earlier workers have also suggested selection of parents from diverse clusters for utilization in hybridization programme to obtain desirable transgressive segregants [41, 42]. In the present study, the lowest inter-cluster distance was recorded between cluster II and III, indicated the genotypes belonging to each pair of the cluster were less diverse or there is close relationship among the genotypes included in these clusters.

Considering the limitations of the morphological characterization in order to have a clear-cut estimation of genetic diversity, SSR markers were used to estimate the genetic diversity and to discriminate genotypes from different regions of the countries. Molecular markers have been the method of interest for the estimation of genetic diversity in field pea [43].

SSR markers produced 25 alleles among the 56 garden pea genotypes, and the average values of the Na (number of alleles), Ne (effective number of alleles), I (Shannon's Information index), PIC (polymorphism information content), Ho (observed heterozygosity) and He (expected heterozygosity) were 3.13, 1.85, 0.71, 0.36, 0.002 and 0.41, respectively. The average value of number of alleles per locus was 3.13 which is consistent with earlier study [6] where an average number of alleles were 3.10 by using 15 SSR based markers in a population of 7 accessions of *P. sativum* L. subsp. *sativum* was reported and lower than the other study [44, 45] where the average number of alleles were 3.6 and 3.8, respectively in pea. The PIC, which represents the allele numbers and their distribution, was figure out to determine the informativeness of each and every marker. The lower PIC value implies a higher level of genetic similarity within the analysed crop genotypes and the *vice-versa*. The most informative SSR marker was AB68, with a PIC value of 0.61, while the minimum PIC was recorded for AB45 (PIC = 0.13). The average PIC value in the present study (0.36) was almost similar with 0.38 for 22 cultivated (*P. sativum)* and two wild relatives (*Pisum fulvum*) reported by Yang et al. [46] and lower than the 0.4817 for 266 grass pea and 17 relative accessions estimated by Wang et al. [47] and the 0.627 for thirty-five pea genotypes estimated by Ahmad et al. [43]. The mean value of expected heterozygosity (0.41) was comparable with 0.43 for 20 grass pea accessions reported by Shiferaw et al. [48]. The polymorphic markers were used for genetic diversity analysis but the observed heterozygosity was low 0.02 which was detected only for locus AA369 and no marker heterozygosity were observed for rest of the locus. Teshome et al. [6] also reported low heterozygosity with the highest value of 0.05. The low heterozygosity in pea is attributable to its highly self-pollinated nature bearing cleistogamous flowers [6].

Estimation of the genetic distance play an important role for effective utilization of the diverse genotypes for hybridization programs [49]. The narrowest pair wise genetic similarity was found between three pairs, i.e. 0.33, with this high divergence/ dissimilarity, these pairs could be used in further breeding programs to developing new segregants. On the other hand, highest genetic similarity was observed between 47 pairs (1.00), suggesting that genotypes of these three pairs share a common genetic background. Such pairs, for having the same similarity standards, are not recommended for use in breeding program, avoiding restriction in the genetic variability, in order to derail the gain to be obtained by selection. The average genetic distance recorded in this study (0.76) is higher than the earlier report by Cupic et al. [50] they reported the range of genetic distance from 0.24 to 0.84. Cluster analysis is one of the most important methods for breeders to know the relationships and genetic diversity of parents for

efficient hybridization program [49, 51]. In the present study, the 56 garden pea genotypes divided into two major group by using SSR markers (Fig 5). The clustering of the genotypes indicated no parallelism between genetic diversity and geographical diversity, since the genotypes of various geographic regions were grouped in different clusters. Similar results were reported in pea by earlier workers [52] who reported that 35 pea accessions were grouped into two major clusters by using 15 polymorphic SSR markers, Wang et al. [47] found that 10 species of Lathyrus genus clustered into two major groups, Arslan et al. [53] have found two major groups of 22 grass pea genotypes. The result of cluster analysis based on the morphological and molecular markers was not similar and it may be due to the environmental influence on the morphological traits.

Molecular variation among garden pea genotypes was estimated by calculating molecular analysis of variance (AMOVA). The results of the study indicated that the high proportion (94%) of variation was due to differences within population, while 6% was due to differences among the population. Liu et al. [54] also reported the use of AMOVA in the genetic diversity study of garden pea and observed 68.14% variation within groups and species while there was 18.33% variation among species and 13.53% variation among groups within species. PCoA is one of the multivariate approaches for grouping of the genotypes based on similarity coefficients which provide more important information about major groups in comparison to the cluster analysis. Fifty-six garden pea genotypes divided into four group by PCoA which is consistence with results obtained by Zhu et al. [55] for 165 cauliflower inbred lines.

Population structure analysis revealed four sub-populations in the 56 garden pea genotypes by using the model-based method STRUCTURE. The similar results were found by Zhu et al. [55] for 165 cauliflower inbred lines. It was also observed that the structure analysis did not evince a clear classification pattern of most of the pea genotypes according to their geographical region, maturity and pod yield. For example, the garden pea genotypes derived from CSKHPKV, Palampur and IIVR, Varanasi under AICRP were grouped into all four sub-populations. This can probably be attributed to the rapid seed dissemination by brids and seed exchanges between farmers or migrate by human intervention to different regions for cultivation point of view. Some discrepancies were also observed between cluster and structure analysis, as cluster analysis assigned a certain branch position for each genotype, whereas structure analysis divided the individuals into sub-populations [35].

Maturity and pod yield is a highly important traits that are considered by breeders for the classification of garden pea cultivars. However, we observed that most of the garden pea genotypes did not cluster as per their maturity as well as their pod yield (Fig 5). For example, group A included early, intermediate and late maturing genotypes and high, intermediate and low pod yielding garden pea genotypes together. The results indicated the presence of some introgression into the gene pool of garden pea genotypes belongs to the different maturity and pod yield group. Therefore, in garden pea breeding program, genotypes having different maturity time and pod yield should be utilized expansively to get desirable hybridization combinations.

## Conclusions

In the current project, a multidisciplinary strategy was carried out with the objective of estimating the genetic variability among the garden pea collection. The studied agro-morphological and molecular markers detected good genetic variability among the 56 *Pisum sativum* genotypes, potentiating their use in garden pea breeding program. SSR markers grouped the genotypes into two major categories by cluster analysis and grouped into four sub-populations by structure analysis, which suggested that the classification of the genotypes was not consistent with their geographical region, maturity and pod yield. The genotypes pair AP-0.3–129

and 2018/PMVAR/5; pair AP-0.3–129 and 2019/PMVAR/1, and pair L-50-1113-1 and 2019/PMVAR/1 exhibited the greatest dissimilarity based on SSR analysis, therefore, these pairs could be used in further pea breeding program to developing new segregants. The proposed combined approaches of morpho-agronomic characterization together with a molecular evaluation in our study can be useful to select diverse parental lines and widen gene-pool of garden pea for future breeding programs.

## Supporting information

**S1 Table. The garden pea genotypes evaluated for SSR polymorphisms in this study.**
(DOCX)

**S2 Table. The sequences of the polymorphic SSR markers.**
(DOCX)

**S3 Table. Genetic distance matrix data calculated for the 56 garden pea genotypes on the basis of the SSR markers.**
(TXT)

**S4 Table. Evaluation of garden pea genotypes against powdery mildew disease resistance.**
(DOCX)

**S1 Data.**
(DOCX)

**S2 Data.**
(XLSX)

**S3 Data.**
(XLSX)

**S4 Data.**
(XLSX)

**S1 File.**
(DOCX)

**S1 Raw image.**
(PDF)

## Acknowledgments

The authors are grateful Dr. R.C. Agrawal, National Coordinator, National Agriculture Higher Education Project (NAHEP) for constant encouragement during the course of this study. Also, we acknowledge Punjab Agricultural University (PAU), Ludhiana; Indian Agricultural Research Institute (IARI), New Delhi; ICAR-Indian Institute of Vegetable Research (IIVR), Varanasi; ICAR-IARI, Regional Station (RS), Katrain; Chandra Shekhar Azad University of Agriculture & Technology (CSAUA&T), Kanpur for providing garden pea genotypes.

## Author Contributions

**Conceptualization:** Akhilesh Sharma, Parveen Sharma.

**Data curation:** Shimalika Sharma, Parveen Sharma.

**Formal analysis:** Nimit Kumar.

**Funding acquisition:** Akhilesh Sharma, Ranbir Singh Rana.

**Investigation:** Parveen Sharma.

**Methodology:** Akhilesh Sharma, Parveen Sharma.

**Software:** Nimit Kumar.

**Supervision:** Akhilesh Sharma, Parveen Sharma.

**Writing – original draft:** Shimalika Sharma.

**Writing – review & editing:** Nimit Kumar, Ranbir Singh Rana, Parveen Sharma, Prabhat Kumar, Menisha Rani.

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
