## [Decision Letter · Decision Letter 0]

22 Apr 2022

PONE-D-22-00070Morpho-molecular genetic diversity and population structure analysis in garden pea (Pisum sativum L.) genotypes using simple sequence repeat markersPLOS ONE

Dear Dr. Sharma,

Thank you for submitting your manuscript to PLOS ONE. After careful consideration, we feel that it has merit but does not fully meet PLOS ONE’s publication criteria as it currently stands. Therefore, we invite you to submit a revised version of the manuscript that addresses the points raised during the review process.

We look forward to receiving your revised manuscript.

Kind regards,

Mohammad Ansari

Academic Editor

PLOS ONE

Journal Requirements:

The authors are grateful to the National Coordinator, NAHEP-ICAR, New Delhi, and the Project Coordinator, CAAST-NAHEP, ICAR, New Delhi for financial support during the study.

Reviewers' comments:

Reviewer's Responses to Questions

**Comments to the Author**

1. Is the manuscript technically sound, and do the data support the conclusions?

Reviewer #1: Yes

Reviewer #2: Yes

Reviewer #3: No

2. Has the statistical analysis been performed appropriately and rigorously? 

Reviewer #1: Yes

Reviewer #2: Yes

Reviewer #3: Yes

3. Have the authors made all data underlying the findings in their manuscript fully available?

Reviewer #1: Yes

Reviewer #2: Yes

Reviewer #3: Yes

4. Is the manuscript presented in an intelligible fashion and written in standard English?

Reviewer #1: Yes

Reviewer #2: Yes

Reviewer #3: No

5. Review Comments to the Author

Reviewer #1: Indeed the article is good but the authors must define

1- The innovation in the study with previous studies.

2- The future prospect and output of the study in which the study is based.

3- What is the take home message of the study. That means any genotype most promising or least promising for future studies and why?

Reviewer #2: The manuscript entitled “Morpho-molecular genetic diversity and population structure analysis in garden pea (Pisum sativum L.) genotypes using simple sequence repeat markers" by Sharma et al. is well written and organized.

Herein, the authors have demonstrated the genetic diversity among 56 garden pea genotypes using morphological descriptors, quantitative traits, and simple sequence repeat markers. I feel that the results provided by the authors will contribute to progress in the future garden pea breeding programs. As such, the manuscript contains a large amount of data. In general, the paper presents results that are novel and merits publication.

So, I strongly recommend the manuscript and can be accepted for publication after incorporating corrections as suggested below:

1. Figures 3 and 6 are not clearly visible.

Reviewer #3: Study by sharma et al on Morpho-molecular genetic diversity and population structure analysis in garden pea genotypes using SSR markers presents a very general overview of estimating genetic diversity. While the presentation and the text of the manuscript needs substantial improvement, my major concern is the rationale and relevance of the such study. In the era of omics, and with a better understanding of phenotype in during the past decade (a product of genotype x environment interaction), there could be opportunities to test this similar set of genotypes for specific phenotype and associated markers. Authors conclude their results with narrow genetic diversity among the genotypes and suggest gene pyramiding to enhance the genetic diversity? I could not understand it. Why not mutation breeding, which is a proven strategy to induce diversity in the such scenario. Overall, this study looks highly repetitive with the previous ones (that authors have mentioned frequently in the discussion section for e.g. page 17, line 313-315; line 318; page 18-19 line 358-362 and many more places) and doesn't add anything new to the existing information.

The sentences are often vague and at places incomplete (line 294). The presentation of results is complex while the outcome of the study is very generalised. Authors did not discuss, why the panel was chosen and what are the reasons or limitations with marker analysis to see the genetic diversity. Also, why morphological markers were chosen in study when it was clear at the first place that they are highly variable and do not help in genetic diversity analysis? these points needs very strong justification to make this article interesting for the readers. I can not recommend this manuscript for publication in its present form.

6. PLOS authors have the option to publish the peer review history of their article (what does this mean?). If published, this will include your full peer review and any attached files.

Reviewer #1: No

Reviewer #2: No

Reviewer #3: No

---

## [Author Response · Author response to Decision Letter 0]

8 Jul 2022

Sub and Reference: Point wise responses to the comments by the academic editor/ reviewers

I have revised our manuscript (PONE-D-22-00070) as per comments and suggestions raised by Academic Editor and Reviewer#1, Reviewer#2 and Reviewer#3. I used the Track Changes for revision to highlight the additions/deletions/changes. Point wise responses to the comments/ suggestions are as below:

Academic Editor:

General 1: After careful consideration, we feel that it has merit but does not fully meet PLOS ONE’s publication criteria as it currently stands. Therefore, we invite you to submit a revised version of the manuscript that addresses the points raised during the review process.

Response: Dear Sir, I have revised the manuscript as per the comments and suggestions. I am greatly thankful to you for positive decision on this manuscript.

Reviewer#1

Comment 1: The innovation in the study with previous studies.

Response: In the current study, we compared the combination of agro-morphological approach together with a molecular evaluation in order to gain a better knowledge on available garden pea genotypes. As results which we got from the project, both stages of characterization are crucial for a better understanding of the genetic variability.

 In the present study, we also screened the genotypes against powdery mildew disease, and reported diverse genotypes (susceptible and resistant based on cluster analysis) for development of mapping population.

 Addition at: Line 32-36, 75-79, 85-87, 353-360

Comments 2: The future prospect and output of the study in which the study is based

Response: Presented in discussion and conclusion section.

 Line 379-383, 461-467

Comments 3: What is the take home message of the study. That means any genotype most promising or least promising for future studies and why?

Response: Presented at Line 345-365, 379-383, 398-404, 451-454, 461-467

Reviewer#2

General: The manuscript entitled “Morpho-molecular genetic diversity and population structure analysis in garden pea (Pisum sativum L.) genotypes using simple sequence repeat markers" by Sharma et al. is well written and organized. I feel that the results provided by the authors will contribute to progress in the future garden pea breeding programs. As such, the manuscript contains a large amount of data. In general, the paper presents results that are novel and merits publication. So, I strongly recommend the manuscript and can be accepted for publication

Response: I am greatly thankful to the reviewer to recommending our research for publication in reputed journal “PLoS ONE”.

Comment 1: Figures 3 and 6 are not clearly visible.

Response: Added new figure with clear visibility.

Reviewer#3

General comment 1: My major concern is the rationale and relevance of the such study.

Response: In the current study, we compared the combination of agro-morphological approach together with a molecular evaluation in order to gain a better knowledge on available garden pea genotypes. Meanwhile, information on characterization of garden pea genotypes using combination of agro-morphological approach together with a molecular evaluation is scare.

General comment 2: Overall, this study looks highly repetitive with the previous ones (that authors have mentioned frequently in the discussion section for e.g. page 17, line 313-315; line 318; page 18-19 line 358-362 and many more places)

Response: The cited studied to support our results were on diversity analysis using agronomic traits or some of them use molecular approach. Some of them are on other crops like Cauliflower or Maize to better support of statements. There are limited studies on both approaches i.e. agro-morphological along with molecular evaluation.

General comment 3: The sentences are often vague and at places incomplete (line 294).

Response: Corrected all incomplete sentences throughout the manuscript.

General comment 4: The presentation of results is complex while the outcome of the study is very generalised.

Response: Added outcome of the study in discussion section as well under conclusion part.

General comment 5: Also, why morphological markers were chosen in study when it was clear at the first place that they are highly variable and do not help in genetic diversity analysis?

Response: We stated that both the strategies are useful for diversity analysis. Because when, we compared the groups formed by agro-morphological and molecular data, no genotypes were observed, indicating that both stages of characterization are crucial for a better understanding of the genetic variability.

---

## [Decision Letter · Decision Letter 1]

10 Aug 2022

Morpho-molecular genetic diversity and population structure analysis in garden pea (Pisum sativum L.) genotypes using simple sequence repeat markers

PONE-D-22-00070R1

Dear Sir

We’re pleased to inform you that your manuscript has been judged scientifically suitable for publication and will be formally accepted for publication once it meets all outstanding technical requirements.

Kind regards,

Mohar Singh, Ph.D. FNASc.

Academic Editor

PLOS ONE

Additional Editor Comments (optional):

Congratulations, i have accepted your publication

Reviewers' comments:

Reviewer's Responses to Questions

**Comments to the Author**

1. If the authors have adequately addressed your comments raised in a previous round of review and you feel that this manuscript is now acceptable for publication, you may indicate that here to bypass the “Comments to the Author” section, enter your conflict of interest statement in the “Confidential to Editor” section, and submit your "Accept" recommendation.

Reviewer #1: All comments have been addressed

Reviewer #3: All comments have been addressed

2. Is the manuscript technically sound, and do the data support the conclusions?

Reviewer #1: Yes

Reviewer #3: Yes

3. Has the statistical analysis been performed appropriately and rigorously? 

Reviewer #1: Yes

Reviewer #3: Yes

4. Have the authors made all data underlying the findings in their manuscript fully available?

Reviewer #1: Yes

Reviewer #3: Yes

5. Is the manuscript presented in an intelligible fashion and written in standard English?

Reviewer #1: Yes

Reviewer #3: Yes

6. Review Comments to the Author

Reviewer #1: I accept all the comments made by the reviewers. The reviewers have revised the manuscript as per the reviewers comments and now the manuscript is in perfect shape.

Reviewer #3: It could be better if the authors have shown the changes marked in the manuscript. Just wrting in response that we have addressed the comments doesn't look enough. No further comment on manuscript.

7. PLOS authors have the option to publish the peer review history of their article (what does this mean?). If published, this will include your full peer review and any attached files.

Reviewer #1: No

Reviewer #3: No

---

## [Editor Report · Acceptance letter]

7 Sep 2022

PONE-D-22-00070R1 

Morpho-molecular genetic diversity and population structure analysis in garden pea (*Pisum sativum* L.) genotypes using simple sequence repeat markers 

Dear Dr. Sharma:

I'm pleased to inform you that your manuscript has been deemed suitable for publication in PLOS ONE. Congratulations! Your manuscript is now with our production department. 

Kind regards, 

on behalf of

Dr. Mohar Singh 

Academic Editor

PLOS ONE